# Haemonchosis in Sheep and Goats, Control Strategies and Development of Vaccines against *Haemonchus contortus*

**DOI:** 10.3390/ani12182339

**Published:** 2022-09-08

**Authors:** Isabella Adduci, Floriana Sajovitz, Barbara Hinney, Katharina Lichtmannsperger, Anja Joachim, Thomas Wittek, Shi Yan

**Affiliations:** 1Institute of Parasitology, Department of Pathobiology, University of Veterinary Medicine Vienna, Veterinärplatz 1, A-1210 Wien, Austria; 2University Clinic for Ruminants, Department for Farm Animals and Veterinary Public Health, University of Veterinary Medicine Vienna, Veterinärplatz 1, A-1210 Wien, Austria

**Keywords:** nematodes, *Haemonchus contortus*, antigens, immune responses, vaccine

## Abstract

**Simple Summary:**

*Haemonchus contortus* is the most pathogenic blood-feeding parasitic nematode in sheep and goats, threatening animal welfare and causing tremendous economic losses to the small ruminant industry. This comprehensive review article sums up current control strategies, worm-derived antigens and recent advances in anti-*Haemonchus* vaccine development. New insights into antigen engineering and general considerations for clinical trials are discussed here.

**Abstract:**

The evolutionary success of parasitic worms causes significant economic losses and animal health problems, including in the small ruminant industry. The hematophagous nematode *Haemonchus contortus* is a common endoparasite that infects wild and domestic ruminants worldwide, especially in tropical and subtropical regions. To date, the most commonly applied control strategy is the administration of anthelminthic drugs. The main disadvantages of these chemicals are their ecotoxic effects, the necessary withdrawal period (especially important in dairy animals) and the increasing development of resistance. Vaccines offer an attractive alternative control strategy against *Haemonchus* infections. In previous years, several potential vaccine antigens prepared from *H. contortus* using the latest technologies have been assessed in clinical trials using different methods and strategies. This review highlights the current state of knowledge on anti-*H. contortus* vaccines (covering native, recombinant and DNA-based vaccines), including an evaluation, as well a discussion of the challenges and achievements in developing protective, efficient, and long-lasting vaccines to control *H. contortus* infection and haemonchosis in small ruminants. This paper also addresses novel developments tackling the challenge of glycosylation of putative candidates in recombinant form.

## 1. Introduction

Among endoparasitic helminths, *Haemonchus contortus*, known by its trivial name ‘barber’s pole worm’, is one of the most important parasites that infects small ruminants and causes major losses to the livestock industry worldwide [1,2]. According to the FAO, around 1.13 billion goats and 1.26 billion sheep were kept worldwide in 2020, indicating an over 20% increase of animal numbers in comparison to 2010. India, China, Nigeria, Pakistan and Bangladesh have the largest stocks of goats, whereas China, India, Australia, Nigeria and Iran represent the top five countries for sheep [3].

*Haemonchus contortus* is a blood-sucking nematode that feeds on blood from capillaries in the abomasum of ruminants [4]. As a single worm ingests up to 50 µL of blood per day, high infection levels can cause severe blood loss (more than 100 ml daily), followed by anaemia and hypoproteinaemia. *H. contortus*-infected animals tend to have a reduced digestive capacity, which affects the uptake of nitrogen, organic matter and energy. In cases of heavy infection, animal death may occur [5,6,7].

Currently, control strategies against haemonchosis are mainly based on the application of anthelmintic drugs developed in the second half of the twentieth century. However, the widespread use of such drugs over the past few decades has resulted in increasing resistance in different parasitic worms, becoming an emerging issue worldwide [5,8,9]. The rapid growth in anthelmintic resistance in *H. contortus* reported in sheep farms globally has dramatically increased the need for alternative and sustainable control strategies [10].

Vaccination is considered a sustainable and efficient option to control infectious diseases including parasitoses. However, developing safe and efficient vaccines against multicellular parasitic worms requires a deep understanding of the biology of both parasite and hosts and of the biochemical properties of parasite-derived molecules, and needs appropriate tools to assess host immune responses in efficacy studies in natural host species, which makes the task rather challenging. As a result, and in contrast to available vaccines against viral and bacterial pathogens, only a handful vaccines against helminth infections are available for livestock, whereas none is currently available for human use [11]. Given the biological nature of *H. contortus*, i.e., with a large part of the population not being in touch with the host immune system and with a high reproductive capacity, the major goal of developing anti-*Haemonchus* vaccines is to decrease worm burden in host animals, rather than eradicating the parasite. Hence, vaccination of both sheep and goats should not only prevent severe diseases caused *H. contortus*, but also reduce transmission via contaminated pastures, which occurs frequently because of the high amounts of eggs shed by infected individuals [12].

In previous decades, immunisation using several different vaccine preparations of small ruminants with *H. contortus* antigens was assessed [13]. For instance, vaccination with native antigens including membrane proteins expressed on the microvillar surface of *Haemonchus* intestinal cells resulted in a partial reduction of faecal egg production and worm burden [14,15,16]. Data have shown that different degrees of protection against *H. contortus* was achieved after vaccination with different native proteins [17]. These early studies successfully identified a group of potent vaccine candidates, which were also assessed in their recombinant forms in follow-up studies [18].

## 2. *Haemonchus contortus*

*Haemonchus* has a two-phase life cycle: a free-living (eggs and larvae L1 to L3) and a parasitic period within the abomasum of the host as larvae (L4 and L5) and adult worms (for details see Figure 1) [19]. The infectious larvae (L3) show a high resistance and survivability under suitable environmental conditions, i.e., high humidity and warm temperature; the parasite is widely found in tropical and sub-tropical regions. However, global warming seems to be causing the spread of *H. contortus* to temperate regions like northern Europe [20,21]. It has been proposed that seasonal changes, primarily in temperature and humidity, and/or hormonal changes are the major determinants for hypobiosis of the L4 [22,23].

*Haemonchus contortus* predominantly parasitises sheep and goats, but is also capable of establishing infection in different breeds of domestic cattle, albeit with lower pathogenicity [24,25,26]. Like small ruminants, new world camels (llamas and alpacas) are susceptible to *H. contortus*, and in these species, the nematode resides the third gastric compartment [27].

Sheep and goats infected with blood-sucking *H. contortus* may show anaemia, eosinophilia and hypoproteinaemia, resulting in pale mucous membranes and submandibular oedema (so-called “bottle jaw”; see Figure 2) [6,28]. Haemonchosis occurs mainly in non-immune young animals (i.e., during first grazing season) or in adult sheep with a weakened immune system [29]. While diarrhoea is a common symptom of intestinal nematode infections, the faeces of *Haemonchus*-infected animals tend to be firm and may appear to be darker due to the occult blood [6].

Post-mortem, a general pallor of the carcass and ascites may be noticeable [6,27]. The abomasal mucosa appears oedematous and congested and exhibits petechial haemorrhages, mainly in the fundus region [30]. Especially in acute haemonchosis, *H. contortus* is readily visible on the surface of the abomasum as a round worm of 2 cm in length with the typical red and white spiral appearance (‘barber’s pool worm) of the females [6]. In the *lamina propria* and submucosa, oedema, congestion as well as hyperplasia of mucus-producing cells can be seen histopathologically [30,31]. An accumulation of inflammatory cells such as eosinophils, lymphocytes and plasma cells is more pronounced in the *lamina propria* than in the submucosa [30].

## 3. Current Control Strategies

Most control strategies to prevent and control haemonchosis do not especially target *H. contortus*, but rather, trichostrongyloids and other gastrointestinal nematodes (GIN) in general. The overuse (and misuse) of anthelmintics has resulted in anthelmintic resistance (AR) of trichostrongyloids and *H. contortus*, in particular against all groups of anthelmintics that are currently on the market [6,32,33,34]. The development of AR threatens small ruminant farming worldwide. In cases of complete anthelmintic drug failure, keeping livestock on grassland becomes uneconomic or even completely unsustainable in most production systems [2,5,9,10,35].

Infections should ideally be controlled in such a way that animal health and productivity are maintained without the promotion of AR development. This can be achieved with integrated parasite management (IPM) concepts that combine chemical and non-chemical strategies [6,10,33,34]. Not all elements of IPM are useful or feasible on each farm and it can be challenging to give and apply best practice advice. However, an increasing amount of recommendations on how to combine different control and treatment strategies are available or under development [10,33,36,37,38].

Strategies which optimise anthelmintic efficacy of compounds in cases where lack of efficacy has already been observed have been developed [39], but it is unclear for how long these strategies will delay the event of complete anthelmintic drug failure. Besides anthelmintics (AH), also alternative compounds with anthelmintic activity, such as bioactive phytochemicals, are available. They have been shown to reduce worm burden and can thus reduce the number of necessary AH treatments [10,34,39].

Non-chemical strategies of parasite control are essential parts of IPM as they reduce the dependency from AH. They can roughly be classified into pasture management and strategies that aim to improve host immune responses against parasite infections through resilience (the ability of the host to withstand the pathological effects of worm-infection) and resistance (the ability of the host to control the intestinal worm burden) [34]. Strategies that aim to improve host responses against parasite infections are nutritional supplementation, genetic selection and vaccination [34]. In general, optimised nutrition is one key factor for resilience and resistance against GIN-infections. Especially supplementary feeding of protein has been shown to enhance the host’s immune response against *H. contortus* infections [34,40,41,42]. Genetic selection for traits that lead to host resistance and/or resilience is a further promising strategy of IPM, as such traits are often highly hereditary [10,34,43,44,45]. Vaccination against haemonchosis offers a further perspective for parasite control (see Section 4). Promising results show its ability to induce a protective host response [18], which is already utilised for farm systems were multi-drug resistance is present [12]. Strategies for how to combine vaccination with elements of IPM will need to be thoroughly investigated in the future.

## 4. Vaccination against *H. contortus*

The development of vaccines targeting parasitic helminths began in the 1950s. The immunisation of calves with radiation-attenuated lungworm larvae (*Dictyocaulus viviparus*) provided a high level of protection against infection. This preliminary work laid the foundation of producing the live vaccine Dictol (Bovilis^®^ Huskvac, MSD Animal Health, Ireland), which was the only commercially available vaccine for a parasitic ruminant nematode at the time [18].

With the relentless increase of anthelmintic resistance, the development of an effective anti-haemonchosis vaccine offers major additional challenges for researchers. Although *H. contortus* is one of the most studied nematodes, the biochemical properties and biological function of parasite-derived molecules remained unknown until 10 years ago. Certainly, the availability of *H. contortus* genomes as well as sophisticated proteomic and transcriptomic studies provided insights about the key players in parasitic and free life stages [46]. One of the crucial problems is still represented by the incomplete immunological knowledge of host–parasite interactions and the immune mechanisms that confer protection against *Haemonchus* [47]. Despite all of these obstacles, important progresses in identifying different *H. contortus* antigens suitable for the development of an effective vaccine have been made in the past 30 years [18].

An early study indicated that oral administration of X-ray attenuated *H. contortus* infectious larvae to sheep could result in protection in mature ewes, despite this method failed to produce reliable levels of protection in three-months old lambs [48]. Since the late 1980s, vaccine development began to focus on identifying and evaluating immunogenic molecules present in fractionated worm lysate and excretory-secretory (ES) products of adult parasites, which are conventional antigens that can be recognised by antisera of naturally infected sheep and goats. In addition, glycoproteins isolated from larval cuticle surface and from the digestive track of adult parasites became a new group of attractive vaccine targets [49,50].

Among many native and recombinant antigens tested in the past, most targets were identified using L3 and adult worms (Figure 1). Recently, bioinformatics analyses of *H. contortus* proteome data provided the scientific community a systematic view of protein profiles during parasite development and the information regarding which proteins are predominant at host–parasite interaction interphase [51,52]. These studies revealed novel potential targets for the control of haemonchosis.

### 4.1. Native Antigen Vaccine

Since the early 1990s, several vaccine candidates, including protein-based antigens, recombinant antigens and DNA vaccines have been identified and tested against the blood-feeding nematode, *H. contortus*. Data show that native proteins extracted from the adult worms gut or from excretory-secretory (ES) products (Table 1) are able to induce high or moderate levels of protective immunity [53].

Native antigens used for vaccine trials could be grouped as either natural or hidden antigens. Natural antigens represent conventional surface antigens and ES products. Hidden antigens represent antigens localised in the digestive tract of the nematodes, which are normally not directly exposed to the host immune system thus do not cross-react with anti-sera of natural infections. However, hidden antigens can efficiently induce antigen-specific antibodies when they are used as immunogens. Because *Haemonchus* feeds on blood, hidden antigens can be targeted by the vaccine-induced antibodies. This mechanism is considered promising since, theoretically, it is capable of limiting the selective pressure to evade the immune response [75].

It has been shown that a moderate protective immune response can be achieved in sheep receiving a purified antigen (*Hc*-sL3) that is expressed on the surface of exsheathed 3rd-stage larvae [56]. Few years earlier, the research group had showed this antigen was an approximately 70–90 kDa protein specific to the larval stage of *H. contortus*. This surface antigen was purified by size exclusion chromatography and was shown to be capable of triggering a local IgA immune response in tickle-infected sheep [76]. Among different the trial groups, sheep that received *Hc*-sL3 and aluminium hydroxide displayed a moderate faecal egg count (FEC) reduction [56].

In another trial lambs were immunised using as natural antigen the somatic protein from adult *H. contortus* named Hc23. The native somatic Hc23 protein was purified previously from the somatic p26/23 fraction isolated from adult *H. contortus* [77]. From this fraction the most abundant protein present was isolated by affinity chromatography and purified by immunochromatography, leading to and allowed the identification of a mass of about 46 kDa [78]. The results suggest that the Hc23 antigen is capable of inducing a partially protective response in lambs [60]. Final data indicated a significant reduction in FECs up to 85.64% and up to 86.00% of reduction of worm counts.

One of the principal protective hidden antigens of *H. contortus* is H11, a group of integral membrane glycoproteins expressed on the nematode’s intestinal microvilli, detected as a major band at 110 kDa in SDS-PAGE under reducing conditions [79]. It belongs to a family of aminopeptidases, and five isoforms named H11-1 to H11-5 have been described by far [66]. Vaccination in sheep with native H11 showed up to 90% reduction in FECs and more than 80% reduction in worm burden (72% for female and 82% for male worms) [55].

Another vaccine trial was performed in lambs vaccinated with a membrane protein extracts from adult worms enriched for cysteine protease activity using thiol-sepharose chromatography. The proteins obtained named TSBP is considered an hidden antigen due to its localisation in the microvillar surface of the intestinal cells of *H. contortus* [58]. SDS-PAGE analyses showed an intricate pattern of bands plus one extra band visible at 60 kDa under reducing conditions. In the experiment conducted under non-reducing conditions, a 60 kDa and more than 250 kDa bands were observed. Moreover, some components of TSBP have been shown to be glycosylated. Like native H11 and H-gal-GP antigens, TSBP has also been shown to contain protease activity [80]. Three vaccination trials showed that lambs immunised with TSBP achieved up to 77% reduction of faecal egg outputs and halving of the final worm burdens [58].

Previous studies first identified a galactose-containing glycoprotein fraction of *H. contortus* which is considered a hidden protective antigen [57,81]. Of these, H-gal-GP is a membrane-associated multi-enzyme protease complex and is involved in blood meal degradation [82]. Most recent structural analysis showed that the complex is capable of cleaving the major components of blood, haemoglobin and serum albumin, into 20-mer peptides and transport them across the membrane [83]. Vaccination with H-gal-GP reached up to 93% reduction of FECs and up to 72% reduction in worm burden, which indicated a lower efficacy in comparison to native H11 antigens [57].

In 2014, the commercial product Barbervax^®^ (Wormvax, Australia), a vaccine containing the two hidden antigens H11 and H-gal-GP complex, was licensed in Australia for the use in sheep. The native H11 glycoproteins are primarily aminopeptidases as well as a small portion of soluble proteins co-purified from the intestinal fraction of the parasite, whereas H-gal-GP is a multi-enzyme complex consisting of a group of aspartyl proteases, metalloproteases and cysteine proteinases. Both antigens may induce strong protective immunity in sheep and goats when used separately. Vaccine trials of Barbervax^®^ have confirmed elevated serum immunoglobulin levels and significant reductions of faecal egg count and worm burden in vaccinated animals [40,59,84,85,86]. Currently, native H11 and H-gal-GP antigens have to be isolated from *H. contortus* adults derived from the abomasa of donor sheep upon slaughter, since adult parasites cannot be obtained by in vitro cultivation. The cumbersome and not very cost-effective procedure is considered a major disadvantage.

Despite the good efficiency achieved during vaccine trials with both native hidden intestinal and somatic antigens of *H. contortus*, their production on industrial scale remains challenging [37]. Since the components of Barbervax^®^ are two hidden native antigens, and repeated doses are required to achieve high levels of protective antibodies, alternative vaccines are required. Therefore, most subsequent attempts shifted to the development of vaccine antigens in their recombinant forms [18].

### 4.2. Recombinant Protein-Based Vaccines

The development of the genetic engineering technologies and immunological proteomics analysis triggered the increasing emergence of the recombinant protein-based vaccines (rPBVs). Therefore, recombinant PBVs are one of the most promising and powerful technologies for vaccine design and artificially inducing immunity [87]. In addition, these vaccines with their relatively cheap production protocols and most importantly, logistical advantages including stability at a wide range of temperatures, could also ensure vaccines access in developing countries.

Recombinant antigens may deliver effective protection comparable to native antigens, and some have been successfully developed into vaccines against metazoan parasites. Although concerning a different phylum, the recombinant Gavac^®^ (Heber Biotec S.A., Cuba) anti-tick vaccine represents a suitable example. It was developed in the 1990s to control *Boophilus microplus* in cattle and it consists of the recombinant Bm86 protein (hidden antigen) expressed in *E. coli* [88]. In addition to ticks, recombinant antigens can also be used to control parasitic nematodes. Due to the complex developmental cycle of parasitic nematodes, to date only a prototype recombinant cocktail vaccine against *Teladorsagia circumcincta* has shown promising results in animal trials. The vaccine contains eight antigens recombinantly expressed either in *E. coli* or in *Pichia pastoris* [89,90].

Regarding *H. contortus*, a number of trials using different recombinant antigens have been reported (Table 1); H11, H-gal-GP, some ES antigens, and the somatic antigens are considered as promising candidates [64,67,91]. Among the trials, somatic antigen Hc23 and ES products ES15/ES24, expressed in *E. coli* and refolded, could significantly reduce the worm burden (70%) in lambs, which is very encouraging, despite the fact that the efficacies are still lower than the aforementioned native hidden antigen H11 [60,61].

Another attempt to immunise sheep in order to prevent infection and produce a significant immune response and protection involved the use of an antigens cocktail, recombinantly expressed, of proteases of H-gal-GP, a well-known protective antigen of *H. contortus*, which is known to confer, in its native version, moderate protection in lambs. Previous data have shown that a major component of the antigen H-gal-GP is a family of at least four zinc metalloendopeptidases, M13 (EC 3.4.24.11), named MEP 1–4. Molecular analysis showed the typical structure of the type II integral membrane proteins regarding MEP 1 and 3; on the other hand, MEP 2 and 4 have putative cleavable signal peptides typical of secreted proteins. MEP1, MEP3 and MEP4 were recombinantly expressed (rMEP1, rMEP3, rMEP4) as soluble proteins using Sf9 insect cells as expression system. For the vaccine trial sheep were immunised both with a recombinant cocktail and the native antigen (as a control group) and developed high levels of serum antibodies, although the protective immunity was only observed in the group vaccinated with native H-gal-GP. A second attempt of using recombinant expression of PEP1 and PEP2, aspartic proteases of *H. contortus*, was unsuccessful. rPEP1 was expressed in *E. coli* and refolded, but the same results as in the previous trial were observed [62,92].

The same strategy of recombinant DNA technology using a different somatic antigen in order to obtain the synthetic protein rHcp26/23 was also evaluated. Despite of the distinct immune response triggered by the vaccination of lambs with the recombinant protein (rHcp26/23), no significant protection against haemonchosis infection after the challenge was reported [63].

Recombinant expression of H11 antigens has been attempted in a few expression hosts, including bacteria, Pichia, Sf9 and Sf21 insect cells as well as the free-living nematode *Caenorhabditis elegans*. Surprisingly, in comparison to the native antigens isolated from adult parasite, none of the recombinant forms could sufficiently reduce worm burden (<30%) and FECs (<40%) in animals post challenge-infection, although anti-H11 antibodies were raised in immunised animals [17,65,66,93]. The inadequate efficacy of recombinant H11s could be due to suboptimal protein folding, lack of other antigenic gut proteins (co-purified with the native proteins) and inappropriate or absent glycan post-translational modifications. Recombinant expression of effective H11 antigens is technically challenging due to the fact that the antigens are naturally glycosylated [94]. Each of the five known H11 isoforms contains multiple N-glycosylation sites, on which the attached glycans seem to be essential for their antigenicity.

Recently, a novel α/β-hydrolase domain protein (HcABHD, a mammalian ABHD17 homolog) was identified in *H. contortus* ES proteins. Data from immunohistochemistry analysis showed a moderate capability of interaction of HcABHD protein with goat T cells in vitro. Results also indicated that recombinant HcABHD was capable of inhibiting secretion of interleukin-4 (IL-4), interferon-gamma (IFN-γ) and transforming growth factor-beta (TGF-β) 1, and elicit the production of IL-10 [95]. Because of the immunomodulatory features during parasite-host interaction, rHcABHD was considered a promising vaccine target. Indeed, administration of recombinant HcABHD in goats led to a moderate reduction of FECs [69]. Another ES antigen, adhesion-regulating molecule 1 (HcADRM1), was expressed in *E. coli* in its soluble form and tested in goats. Results indicated that vaccination with recombinant HcADRM1 induced a slightly protective immunity in comparison to rHcABHD, with reductions of 48.9% in FECs and up to 58.6% in worm burdens [68,96].

All vaccination trials with recombinant antigens against haemonchosis showed similar results, meaning no significant high level of protection compared to the same antigens used in their native forms. The absence or decrease of the protective capacity of recombinant antigens might be related to a lack of post-translational modifications or inaccurate and sub-optimal folding. Given the economic importance of haemonchosis and the widespread of anthelmintic resistance, further investigations must be performed. Indeed, the protein might be expressed in other systems for a better folding and protein glycosylation to allow the proper functioning as a protective antigen. Moreover, different adjuvant systems might be tested in order to increase the immunogenicity.

### 4.3. DNA Vaccines

DNA vaccines represent a relatively new approach to the control of infectious diseases [97]. This technology is used to deliver a genetically modified DNA of a specific antigen to the host so that immune cells can be directly exposed to the antigen and produce a wide range of protective immune responses. Evidence suggests that DNA vaccines have several advantages over conventional vaccines (protein- or subunit-based vaccination methods), including their ability to induce a wider range of immune responses. DNA vaccines are known to be stable at room temperature, which is highly practical for use in endemic and rural regions, relatively cheaper, and the immune response elicited is specific to the encoded protein [98,99].

Moreover, it has been observed that the development of the haemonchosis vaccine requires high levels of antigen-specific antibodies for effective protection, which might be complicated to achieve with this technology [100]. Hence, among several vaccine strategies, DNA vaccines represent an attractive candidate as a tool to control haemonchosis infection in small ruminants. Several studies have been published and showed a partial protection achieved in goats following DNA vaccination (Table 1). However, DNA vaccine trials have only been carried out on goats, so far.

In the first DNA vaccine trial, goats of similar age were immunised with recombinant hidden HC29 DNA vaccine codifying for the enzyme glutathione peroxidase (GPX). Among the essential antioxidant enzymes that are physiologically important for parasites there is glutathione peroxidase (GPX) [101]. The construct named pVAX1/HC29 containing the recombinant hidden antigen seemed to induce a partial immune response in immunised subjects. DNA vaccine conferred 36.1% reductions in FECs and high levels of HC29-specific antibodies (serum IgG, serum and mucosal IgA) and an increase in the CD4^+^ T-lymphocyte population was reported [70].

One year later, a DNA vaccine expressing immunogenic fragments of the hidden antigen H11 with or without IL-2 (Interleukin IL-2) was tested on approximately 10-month-old goats [71]. Following immunisation in the group vaccinated with both H11 and IL-2, high levels of specific immunoglobulin (Ig) G in serum, non-specific IgA in serum, and mucosal IgA were reported. In addition, the presence of CD4+ T lymphocytes, CD8+ T lymphocytes and B lymphocytes was also reported. The vaccine induced partial protection since a 46.7% reduction in abomasal worm burden and 56.6% reduction of FECs was observed.

Two more DNA vaccine candidates with two different *H. contortus* somatic antigens named glyceraldehyde-3-phosphate dehydrogenase (GAPDH) and Dim-1 coupled with recombinant pVAX1 constructs were investigated [72,73]. In both trials, administration was performed in 10-month-old goats. Glyceraldehyde-3-phosphate dehydrogenase (GAPDH) is an important enzyme involved in the energy production, both in glycolysis and gluconeogenesis. Data have also shown its involvement in various parasitic diseases, which is why it is considered to be a promising therapeutic target. A vaccination trial with pVAX1-HcGAPDH showed a 35% and 38% reduction in FEC and abomasal worm burden, respectively; the increasing of antigen-specific IgG and IgA serum levels and CD4+ T-lymphocyte population was also reported [72]. Dim-1 is a structural protein that belongs to the immunoglobulin family. It is considered a potential candidate for a vaccine against haemonchosis. Notably, Dim-1 vaccine was capable of inducing a slightly higher level of protection than GAPDH, showing a reduction of 46% in FECs and 51% in abomasal worm burden [73].

Most recent vaccine trial occurred in 2014, when a DNA vaccine encoding *H. contortus* actin (somatic antigen) was tested for protection against infections in 8 to 10 months old goats. Earlier, the same research group managed to detect an actin homologue in goat serum; this evidence made the actin a potential candidate for vaccination trial. Actin is known to be a globular-shaped protein extremely crucial for the functioning of eukaryotic cells. The highest occurrence of actin occurs in the cells of muscle tissue, where it is essential for the contraction process. Moreover, it is also involved in important cellular processes including cell motility, signalling and cell division [102]. The final data showed that immunised goats showed higher levels of serum IgG, IgA in both serum and mucosal tissue. In addition, an increase in CD4+ T-lymphocytes, CD8+ T-lymphocytes and B-lymphocytes and TGF-β concentrations was reported. Moreover, a reduction of 34.4% and 33.1% in the mean eggs per gram faeces (EPG) and worm burdens in the Actin-vaccinated group compared to the control, was reported in immunised goats [74].

Although high protection against infection has not been fully achieved, all these findings suggest DNA vaccine is a promising approach to induce both specific humoral and cellular immune responses against parasites, and it may offers an alternative strategy to develop cost-effective and highly antigen-specific vaccines against *H. contortus*.

### 4.4. Efficacy Assessment in Clinical Trials

Vaccine efficacy has to be assessed in randomised and controlled clinical trials in order to determine if a vaccine candidate is capable of inducing immune responses in the host (mostly sheep or goats) and to evaluate if the resulted immunity is adequate to suppress experimental infection of third-stage *H. contortus*. A trial normally includes experimental groups, which receive antigens either alone or formulated with an adjuvant to stimulate immune responses, and negative control group(s) that receive only the adjuvant or a buffer. A positive control group that receives a protective antigen, e.g., Barbervax^®^ is not always necessary but becomes relevant when the trial aims to compare efficacies. Clinical trials typically last between 4 to 6 months, including initial diagnostics and deworming steps, immunisation of animals, oral infection with infective larvae and post-mortem examination. During the trial faecal and blood samples are taken from animals and examined using parasitological, immunological, biochemical and haematological methods in addition to monitoring clinical conditions and analysing other specimens. Results of animal trials are often influenced by many factors, such as animal breeds, age, immunisation schedule, antigen dosage, choice of adjuvants and administration route of vaccine et al. Therefore, these factors should be taken into consideration when designing a clinical trial. We selected trials reported in 27 publications, which have relatively complete datasets, and summarised aforementioned factors in Table 2.

The average age of experimental animals seems to influence the outcome of efficacy assessments. For instance, after receiving an equal amount of native ES15/ES24 antigen, 9-month old lambs evoked stronger serum antibody responses than 3-month old lambs, which correlated with the reduction of abomasal worm counts [103]. A similar pattern was observed in another study using whole adult ES products as immunogens [104]. In fact, in most of the reported clinical trials, 6- to 10-month-old lambs were employed (Table 2).

It has been hypothesised that sheep (ovine) and goats (caprine) evolved different strategies to cope up with gastrointestinal nematode infections [105]. Sheep are considered less susceptible to parasitic nematodes than goats and some sheep breeds feature stronger immunities naturally against *H. contortus* than the others [106,107]. In contrast, goats elicit milder immune responses than sheep after receiving vaccine shots, which could be an important reason led to a compromised efficacy [59].

Different adjuvants have been used to formulate anti-*Haemonchus* vaccines in the past, mainly including aluminium-based adjuvant (aluminium hydroxide, Alhydrogel^®^, InvivoGen, San Diego, CA, USA and Rehydragel^®^, Vertellus, Indianapolis, IN, USA), complete and incomplete Freund’s adjuvant, dimethyldioctadecylammonium (DDA) and saponin-based adjuvant (Quil A^®^, InvivoGen, USA). Bacterial lipopolysaccharide (endotoxin) and extracts of insect cell and *C. elegans* were also used as a supplement in some studies, which didn’t seem to significantly increase the efficacy [65,108]. In a study on the effect of adjuvants, aluminium hydroxide mixed with the somatic antigen rHc23 (200 µg/dose) delivered a better efficacy with 71.0% ± 14.3% reduction of worm burden in comparison to Quil A^®^ mixed with the same antigen [109]. Native hidden antigens, H11 and H-gal-GP antigens, were initially mixed with complete and incomplete Freund’s adjuvant and tested in clinical trials in the early 1990s [57,79], which resulted in promising reductions of FECs and worm burden. The high efficacy is reproducible when replacing Freund’s adjuvant with saponin-based adjuvant and the commercially available vaccine Barbervax^®^ contains 1 mg Quil A^®^ [66,110]. Interestingly, aluminium hydroxide preferentially induces Th2 immune responses characterised by IgG1, IgE and eosinophilia, whereas Freund’s adjuvant and saponin induce strong Th1 immune response, featuring high IFN-γ, IgG2 and NK cell activities against intracellular viruses and bacteria infections. Therefore, different immune components may involve in rHc23-induced immunity in comparison to Barbervax^®^-induced immunity.

**Table 2 animals-12-02339-t002:** Data are extracted from selected vaccine trials in literatures. Changes in antibody, FEC and worm burden are relative to the control group in each trial. Abbreviations: NA not applicable, NS no significant differences, ND no data; DDA, dimethyldioctadecylammonium; LPS, lipopolysacharide; i.m., intramuscular injection; s.c., subcutaneous injection. * Antigen is supplied with plasmid DNA encoding caprine IL-2 (100 µg/dose).

Antigens	Adjuvant and Supplement	Animal Breed	Age and Sex	Group Size	Dosage (µg/Dose)	Number of Vaccination	Administration Route *	Infection Dosage L3 Larvae	Efficacy Accessement			
									Antigen-Specific Antibodies	FEC Reduction	Worm Burden Reduction	References
**native or recombiant antigens**												
whole ES antigens	Alhydrogel^®^	Zwartbles lamb	3-month old	5	75	3	s.c.	10,000 to 12,000	IgG↑, IgE↑, IgA↑, IgM↑	89.0%	54.0%	[104]
native p26/23	Freund’s adjuvant	Manchego lamb	3.5 to 5-month ♀	5	50	3	s.c., i.m.	400	IgG↑	64.2%	61.6%	[77]
rHcp26/23	Freund’s adjuvant	Manchego lamb	3-month old ♀	5	100	3	s.c., i.m.	16,000	IgG↑	NS	NS	[63]
native Hc23	aluminium hydroxide	Assaf lamb	4–5 month old ♀	7	100	3	s.c., i.m.	15,000	IgG↑	70.67–85.64%	67.17% to >86%	[60]
rHc23	aluminium hydroxide	Assaf lamb	5–6 month old ♀	7	100	3	NA	15,000	IgG↑	83.5%	84.7%	[64]
rHc23	aluminium hydroxide	Manchego lamb	6-month old ♀	7	200	3	i.m.	4000	IgG↑	82.37 ± 5.98%	71.0 ± 14.3%	[109]
rHc23	Quil A®	Manchego lamb	6-month old ♀	7	200	3	i.m.	4000	IgG↑	74.58 ± 10.94%	47.3 ± 35.4%	
rHc23	LPS	Entrefino lambs	5 to 6-month old ♀	6	100	4	i.m.	6*1000	IgG↑	43.3%	45.5%	[108]
rHc23	LPS	Entrefino lambs	5 to 6-month old ♀	7	100	4	i.m.	6*2000	IgG↑	43.5%	84.3%	
native ES15/ES24	DDA	Texel sheep	8-month old	10	50–100	3	s.c.	20,000	IgG1↑	72.9%	82.2%	[91]
native ES15/ES24	DDA	Texel sheep	3-month old	4	50–100	3	s.c.	10,000	IgG1↑, IgA↑	ND	−34.4%	[103]
native ES15/ES24	DDA	Texel sheep	9-month old	10	50–100	3	s.c.	20,000	IgG1↑, IgA↑	ND	82.2%	
rES15/rES24	DDA/insect cell extract	Texel, Swifter and Zwart Bles	9-month old	7	100 (cocktail)	3	s.c.	5000	IgG1↑	49.0%	65.0%	[61]
native ES15/ES24	DDA	Texel, Swifter and Zwart Bles	9-month old	7	100	3	s.c.	5000	IgG1↑	57.0%	70.0%	
rH11-1	insect cell extract	Merino lamb	5-month old ♀	5	300 (crude extract)	2	i.m	15,000	serum antibody↑	ND	29.0%	[65]
rHcGST-H11-1	insect cell extract	Merino lamb	5-month old ♀	5	300 (crude extract)	2	i.m	15,000	serum antibody↑	ND	20.0%	
native H11	Vax Saponin	Suffolk-cross lamb	6-month old	7	40	3	s.c.	5000	IgG↑, IgE↑, IgM↑	99.9%	93.6%	[66]
rH11-1+rH11-4	Vax Saponin	Suffolk-cross lamb	6-month old	7	20 (cocktail)	3	s.c	5000	IgG↑	NS	NS	
rH11-4/5	Vax Saponin	Suffolk-cross lamb	6-month old	7	20 (cocktail)	3	s.c	5000	IgG↑	NS	NS	
rH11 (1-570 aa)	Freund’s adjuvant	Boer goat	6-month old	5	750 (crude extract)	3	i.m.	5000	IgG↑	37.71%	24.91%	[67]
rH11 (223-570 aa)	Freund’s adjuvant	Boer goat	6-month old	5	50	3	i.m.	5000	IgG↑	26.04%	18.46%	
rH11-5/AC1/PEP1	Quil A/Rehydragel/Covexin	White mountain breed lamb	6-month old	6	100	3	i.m.	5000	IgG↑	23.9%	13.5%	[93]
native H-gal-GP	Quil A®	Suffolk–Greyface crosses	5 to 9-month old	7	100	3	i.m.	5000	IgG↑	93.0%	69.0%	[81]
native H-gal-GP	Freund’s adjuvant	lamb	2-month old	7	200	3	i.m.	5000	ND	93.0%	52.9%	[57]
rMEP-1/3/4/rPEP-1	Quil A®	castrated Blackface×Leicester	9-month old ♂	7	200 (cocktail)	3	i.m.	5000	IgG↑	2.5%	1.0%	[62]
native H-gal-GP	Quil A®	castrated Blackface×Leicester	9-month old ♂	7	100	3	i.m.	5000	IgG↑	88.5%	72.3%	
H11 and H-gal-GP	saponin	crossbred Brergamacia	ewes	39	5 and 50	3 + 5	i.m.	natural infection	IgG↑	NS	ND	[85]
H11 and H-gal-GP	saponin	crossbred Brergamacia	lambs	48	5 and 50	3 + 3	i.m.	natural infection	IgG↑	72.0%	68.0%	
Barbervax®	Quil A®	crossbred Bergamasco	ewes	29	5	3 + 3	s.c.	natural infection	IgG↑	NA	ND	[40]
Barbervax®	Quil A®	crossbred Bergamasco	lambs	29	5	3	s.c.	natural infection	IgG↑	80.0%	ND	
Barbervax®	Quil A®	Saanen Nubian dairy goat	6-month old	10	5	3 + 3	s.c.	3*6000	IgG↑	57.4 ± 17.6%	ND	[59]
Barbervax®	Quil A®	Anglo Nubian dairy goat	6-month old	10	5	3 + 3	s.c.	3*6000	IgG↑	69.8 ± 11.7%	ND	
Barbervax®	Quil A®	alpacas	4 to 6-month old	7	5	3	s.c.	3*1500	IgG↑	NS	ND	[86]
Barbervax®	Quil A®	crossbred Santa Ines hair sheep	ewes	45	5	3 + 5	s.c.	natural infection	IgG↑	90.2 ± 4.03%	ND	[84]
Barbervax®	Quil A®	crossbred Santa Ines hair sheep	1-year old	12	5	3	s.c.	4*3000	IgG↑	87 ± 5.4%	79.0%	
**DNA Vaccine candidates**												
HC29	PBS buffer (pH 7.4)	local-bred goat	8 to 10-month old	5	500	2	i.m	5000	IgG↑	36.1%	35.6%	[70]
H11	PBS buffer (pH 7.4) *	local-bred goat	10-month old	4	300	2	i.m	5000	IgG↑	56.6%	46.7%	[71]
GAPDH	PBS buffer (pH 7.4)	local-bred goat	9 to 10-month old	5	500	2	i.m	5000	IgG↑	34.9%	37.7%	[72]
Dim-1	PBS buffer (pH 7.4)	local-bred goat	8 to 10-month old	5	500	2	i.m.	5000	IgG↑	45.7%	51.1%	[73]
Actin	PBS buffer (pH 7.4)	local-bred goat	8 to 10-month old	5	100	2	i.m.	5000	IgG↑	34.4%	33.1%	[74]

FECs and *post-mortem* worm burden are very likely the two most important evaluation criteria to draw the conclusion if protective immunity can be achieved by administration of candidate vaccines to animals. However, in order to gain more insights into the mechanism of vaccine-induced immunity, it is worth investigating more factors during the trial. Peripheral blood samples, often collected weekly before and after immunisation and challenge infections, contain rich information. Many trials included haematological assessments of packed cell volume (PCV), proportion of lymphocytes (eosinophils and neutrophils). Furthermore, groups of CD4^+^ T cells in the blood can be labelled with specific monoclonal antibodies and measured with Flow cytometry, whereas cytokines levels are often assessed by PCR using reverse-transcribed total RNAs of blood samples.

In all trials, it has been reported that serum antibody titres, especially antigen-specific IgG antibodies, are elevated after vaccination regardless the type of antigen and adjuvant used (Table 2). It is important to emphasise that a high level of antigen-specific IgG may not be sufficient to result in desired protection. For example, lambs immunised with native H11 antigens elicited predominant and persistent IgG antibodies; in addition, low titre IgE and IgM antibodies were also detected in the blood, although they quickly decayed post L3 challenge. In contrast, lambs that received *C. elegans*-derived recombinant antigens solely produced IgG antibodies [66], which were capable of recognising native antigens. Interestingly, a large portion of IgG antibodies actually targeted glycans of H11 proteins in both groups, possibly also cross-reacting with a broader range of parasite glycoproteins modified with the same glycans. This implied the participation of other classes of immunoglobulins (IgA, IgM and IgE), which might have been overlooked in other trials.

*H. contortus* infections induce an unequivocal Th2 immune response in the host. In both sheep and goats, the involvement of IgA and IgE antibodies in natural *H. contortus* infection has been well documented in earlier studies [111,112,113]. Interestingly, the parasite-specific IgA level seems to be higher in *H. contortus* resistant than in susceptible breeds, and there is a negative correlation between the (salivary and mucosal) IgA level and quantitative egg excretion [106,114,115]. As for IgE antibodies, it has been shown that *H. contortus* infection induces parasite-specific IgE antibodies in sheep [116]. Immunisation with ES15/ES24 antigen significantly increased antigen-specific IgE in sera of six to nine months old lambs and the IgE level negatively correlated with worm burden [117]. Intriguingly, serum IgE from *H. contortus*-infected sheep recognises core α1,3-linked fucose, a common glycan epitope present on *N*-glycoprotiens of nematodes [118]. In addition, as mucosal immunity is also considered an important element of natural protective immunity preventing establishment of *H. contortus* in the abomasum [106,119], immunohistochemistry studies of lymphocytes (e.g., neutrophils) and ELISA assays of abomasal antibodies as well as local and systemic cytokine responses should also be intensified in future trials.

## 5. New Perspectives

While the booming -omics studies on *H. contortus* may have accelerated the identification of novel vaccine targets in recent years [51,120,121], we believe that there is room for further improvement of vaccine efficacy of recombinant antigens using protein-engineering approaches. The recombinant production of an effective anti-*Haemonchus* vaccine will overcome the limitations of using native worm antigens, which is within the scope of the 3R rules (reduction, refinement, and replacement of animals) in veterinary medicine.

As many identified *Haemonchus* antigens are N-glycosylated proteins (Table 1), attempts of using unsuitable expression hosts, such as *E. coli*, to produce eukaryotic glycoproteins should not be encouraged in efficacy trials in the future. Instead, the emerging technical trend, i.e., glyco-engineering of eukaryotic expression hosts, should be considered in order to overcome protein folding issues and to achieve desired glycan modifications on recombinant antigens [122]. Engineering of *Nicotiana benthamiana* plants enabled the expression of LDNF and Lewis X glycan epitopes on recombinant egg antigens of *Schistosoma mansoni* (trematode), which significantly promoted Th2 polarisation of dendritic cells (DCs) in a DC-SIGN dependent manner and induced high level of IL-4 secretion in mouse models [123]. Using the same or analogous platforms, it would be technically feasible to engineering the glycosylation of *H. contortus* antigens.

At present, subcutaneous or intramuscular injection are the most common vaccine administration routes in veterinary medicine. The commercially available vaccine Barbervax^®^ was approved for subcutaneous administration in sheep. The skin, as a highly effective component of the immune system, is an attractive target for vaccination due to its high density of immunocompetent cells such as Langerhans cells and dermal dendritic cells that specialise in antigen uptake followed by antigen presentation [124]. In a previous study the skin was utilised and intradermal administration of one-fifth of the full-dose of human influenza antigen revealed in equal immunogenicity as the full-dose intramuscular injection [125]. The use of less amount of antigen needed for intradermal vaccination is so called ‘dose-sparing-effect’. Intradermal administration of purified natural *H. contortus* surface antigens (*Hc*-sL3) on the relatively wool free area of the inner thigh of Merino sheep was feasible and revealed in the reduction of FECs and worm burden [56]. A rabies vaccine was successfully administered by intradermal needle-free vaccination on the dorsal part of the ear leading to protective antibody titres in sheep [126]. However, studies focusing on intradermal vaccine delivery in small ruminants remain limited to date. In the future, the potential of delivering anti-*Haemonchus* vaccines using intradermal administration devices, especially needle-fee devices, should be further explored. 

While trials using monovalent DNA vaccines have achieved low levels of protection in goats (Section 4.3), a multivalent DNA vaccine targeting a group of selected antigens may enhance the efficacy. Furthermore, the mRNA vaccine technology could be exploited to battle parasitic infections in small ruminants. mRNA vaccines, despite its need for cold-chain transport and storage to maintain stability, have shown to be able to stimulate potent immune responses in animals and humans [127,128]. In addition, both DNA and mRNA vaccines are easy to produce in large quantity with relatively low cost, which made them suitable for protecting the large global population of sheep and goats. Therefore, mRNA-based vaccines against *H. contortus* should be attempted in the future.

We are currently in the era of multi-omics, which provides us with sophisticated tools to study host–parasite interactions. Consequentially, more and more novel antigenic targets will be revealed in the near future, which should be experimentally assessed. Unfortunately, currently no clear guideline for the evaluation of anti-*H. contortus* candidate vaccines is available. As aforementioned in Section 4.4, many variables, including animal breed, age, and choice of adjuvant, may influence the outcome of a vaccine trial. It usually takes a few trials until the vaccine formulation of an antigen and its approximate efficacy are optimised. Given the complex nature of host immune responses to administered vaccines and to subsequent parasite challenge, in our opinion, a vaccine trial should not only focus on parasitological parameters such as faecal egg count and worm burden, but also include a broad range of immunological parameters. The desired protective immunity against *H. contortus* is very likely a combination of both humoral and cellular responses [18]. In addition to serum parasite-specific IgG profiles other classes of immunoglobulins also should be characterised, and abomasal immunity, including cytokine profile, mucosal immunoglobulin levels and local cellular immune responses, should be looked at in detail. Although performing such experiments in an efficacy trial adds much more effort and requires expertise in ruminant immunology, it will lead to an overview on how vaccination shapes host immunity and induces protection. Even if the vaccine candidate is considered non-productive in the end, such datasets remain invaluable to the scientific community and will guide future vaccine design.

## Figures and Tables

**Figure 1 animals-12-02339-f001:**
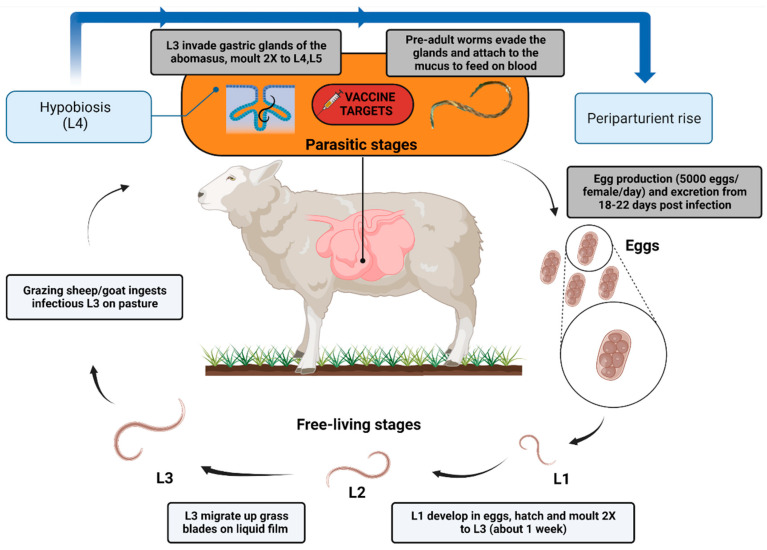
The life cycle of *Haemonchus contortus* in sheep. [Created with BioRender.com, accessed on 21 June 2022].

**Figure 2 animals-12-02339-f002:**
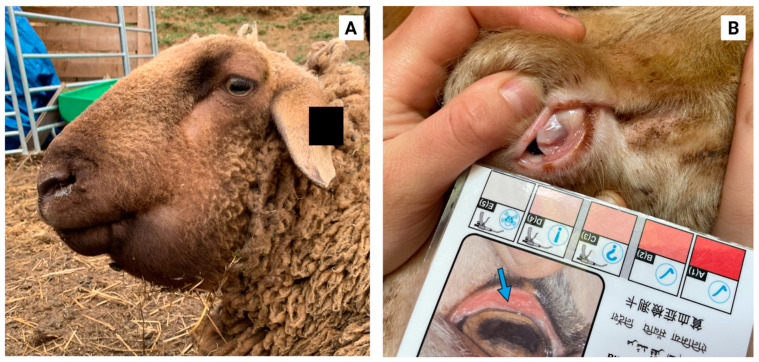
(**A**) An Austrian Jura sheep with typical submandibular oedema (‘bottle jaw’) and (**B**) an East Friesian sheep with pale conjunctiva due to anaemia. The instrument in the picture is a FAMACHA^©^ card in non-English. A FAMACHA^©^ card, a laminated colour chart developed in South Africa, is commonly used for the clinical identification of anaemic sheep and goats.

**Table 1 animals-12-02339-t001:** Vaccine candidates listed here are from protein-based *Haemonchus contortus* antigens that have been tested in host animals and indicated degrees of protection. Their accession numbers, if not mentioned in literature, were determined by performing Nucleotide BLAST search (BLASTn) using reported primer sequences for molecular cloning. N-glycosylation sites were predicted using NetNGlyc—1.0 server [54]. * Defined by the reduction of faecal egg counts compared with adjuvant control group; >80% as high, between 50% and 80% as moderate and <50% as low. ** Described as H11-1 in the publication by mistake. BLASTn search with primers, which were used for amplifying the three gene fragments, indicated that the authors actually cloned the H11 encoding gene (GenBank: X94187.1). Abbreviations: NS, no significant differences; NA, not applicable, ND, no data.

Antigens	Molecular Properties					
	Type (H/N)	Accession Number	N-Glycosylation	Expression Host & Solubility	Efficacy in Challenge Studies *	References
**Native antigens**						
native H11	H	NA	yes	adult worms	high	[55]
*Hc*-sL3	N	NA	yes	third-stage larvae	moderate or low	[56]
H-gal-GP	H	NA	yes	adult worms	moderate or high	[57]
TSBP	H	NA	yes	adult worms	moderate	[58]
Barbervax® or Wirevax	H	NA	yes	adult worms	moderate or high	[59]
Hc23	N (somatic)	GenBank: CDJ92660.1	no	adult worms	high	[60]
**Recombinant parasite proteins**						
ES15	N (E/S)	UniProt: O18518	no	*E. coli*, refolded	low	[61]
ES24	N (E/S)	UniProt: O18519	yes	*E. coli*, refolded	low	
MEP1	H	UniProt: O45131	yes	Sf9 insect cell, soluble	NS	[62]
MEP3	H	UniProt: O76751	yes	Sf9 insect cell, soluble	NS	
MEP4	H	UniProt: Q9Y1I4	yes	Sf9 insect cell, soluble	NS	
PEP1	H	UniProt: Q25037	yes	*E. coli*, refolded	NS	
Hcp26/23	N (somatic)	GenBank: CDJ92660.1	no	*E. coli* M15, refolded	NS	[63]
Hc23	N (somatic)	GenBank: CDJ92660.1	no	E. coli BL21(DE3), NA	high	[64]
H11-1	H	GenBank: CAB57357.1	yes	Sf9 insect cell, soluble	NS	[17]
H11-2	H	GenBank: CAB57358.1	yes	Sf9 insect cell, soluble	NS	
H11	H	UniProt: Q10737.2	yes	Sf9 insect cell, soluble	NS	
H11-1	H	GenBank: CAB57357.1	yes	Sf21 insect cell, soluble	ND	[65]
H11-1	H	GenBank: CAB57357.1	yes	*C. elegans*, soluble	NS	[66]
H11-4	H	UniProt: Q967C6	yes	*C. elegans*, soluble	NS	
H11-5	H	UniProt: V5K5H8	yes	*C. elegans*, soluble	NS	
H11 (partial, 1-570 aa)	H	UniProt: Q10737.2	yes	*C. elegans*, soluble	low	[67]
HcADRM1	N (E/S)	UniProt: W6NKS2	no	*E. coli* BL21 (DE3), NA	moderate or low	[68]
HcABHD	N (E/S)	GenBank: CDJ88804.1	no	*E. coli* BL21 (DE3), NA	moderate	[69]
**DNA Vaccine targets**						
HC29	H	UniProt: D0F095	yes	goats	low	[70]
H11 **	H	GenBank: Q10737.2	yes	goats	low	[71]
GAPDH	N (somatic)	UniProt: D9IL10	yes	goats	low	[72]
Dim-1	N (somatic)	GenBank: ADZ24723.1	yes	goats	low	[73]
Actin	N (somatic)	GenBank: CDJ80138.1 or CDJ93106.1	yes	goats	low	[74]

## Data Availability

Not applicable.

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
