# Peer review of "Haemonchosis in Sheep and Goats, Control Strategies and Development of Vaccines against Haemonchus contortus"

_animals, 2022, doi:10.3390/ani12182339_

Round 1

Reviewer 1 Report

The review was interesting to read and well-written. I would like to see some literature cited showing links/if any with the protection against Haemonchus to the IgA or IgE levels

Reviewer 2 Report

Haemonchosis in small ruminants and development of vaccines against Haemonchus contortus have been well reviewed elsewhere, for example in the theme ‘Haemonchus contortus and Haemonchosis – Past, Present and Future Trends’ in Advances in Parasitology, Volume 93. The content summarized in this manuscript, although comprehensive, is not significantly updated compared to published reviews, nor does it provide enough novel perspectives. Therefore, this manuscript has limited guiding significance for related research fields. I recommend rejection.

Reviewer 3 Report

Haemonchosis in sheep and goats and development of vaccines  against Haemonchus contortus

The author made an excellent work gathering and summarizing available information on the topic of vaccines against H. contortus.

Although it is constructed as a narrative review, the authors made a brief description of factors used when selecting data for table 1.

Therefore, it would be recommended that a brief description is given about their search strategy for recovery vaccine-related work and their inclusion/exclusion criteria.

As the review focus on vaccines, I recommend shortening H. contortus description and eliminating section 3 (current control strategies).

Finally, I suggest including a final section where a guideline/recommendations for "ideal" vaccine testing is described. The review and analysis of the research carried out to date, does yield information of positive and recommended scenarios (already identified by the author in some sections). This would greatly increase the value of this paper/review.

minor additional comments are in the file

Round 2

Reviewer 3 Report

Haemonchosis in sheep and goats, control strategies and development of vaccines against Haemonchus contortus

Authors have properly answered all queries and comments from the previous review round. No further comments

Great job!